# Targeting Natural Killer T Cells in Solid Malignancies

**DOI:** 10.3390/cells10061329

**Published:** 2021-05-27

**Authors:** Zewde Ingram, Shriya Madan, Jenoy Merchant, Zakiya Carter, Zen Gordon, Gregory Carey, Tonya J. Webb

**Affiliations:** Department of Microbiology and Immunology, Marlene and Stewart Greenebaum Comprehensive Cancer Center, University of Maryland School of Medicine, Baltimore, MD 21201, USA; ZIngram@som.umaryland.edu (Z.I.); Shriya.Madan@som.umaryland.edu (S.M.); jmerchant@umaryland.edu (J.M.); Zakiya.Carter@som.umaryland.edu (Z.C.); zenzineer@gmail.com (Z.G.); GCarey@som.umaryland.edu (G.C.)

**Keywords:** NKT cells, CD1d, cancer immunotherapy

## Abstract

Natural killer T (NKT) cells are a unique subset of lymphocytes that recognize lipid antigens in the context of the non-classical class I MHC molecule, CD1d, and serve as a link between the innate and adaptive immune system through their expeditious release of cytokines. Whereas NKT have well-established roles in mitigating a number of human diseases, herein, we focus on their role in cancer. NKT cells have been shown to directly and indirectly mediate anti-tumor immunity and manipulating their effector functions can have therapeutic significances in treatment of cancer. In this review, we highlight several therapeutic strategies that have been used to harness the effector functions of NKT cells to target different types of solid tumors. We also discuss several barriers to the successful utilization of NKT cells and summarize effective strategies being developed to harness the unique strengths of this potent population of T cells. Collectively, studies investigating the therapeutic potential of NKT cells serve not only to advance our understanding of this powerful immune cell subset, but also pave the way for future treatments focused on the modulation of NKT cell responses to enhance cancer immunotherapy.

## 1. Introduction

A hallmark of cancerous cells is their ability to evade destruction by the immune system [1]. This is essential for the tumor because the host immune system possesses the potential to eliminate malignancies, and invokes a multi-layered process that can include early recognition events by mediators of innate immunity, followed by the development of a strong and highly specific adaptive immune response. NKT cell-mediated cytokine production leads to the induction of both the innate and adaptive immune responses; therefore, NKT cells have been implicated in the modulation of immune responses to cancer, autoimmunity, infection, allergy, and transplantation (reviewed in [2,3,4,5]). NKT cells have the capacity to mount strong anti-tumor responses and have thus become a major focus in the development of effective cancer immunotherapy. NKT cells comprise a distinct T lymphocyte subset which display innate effector functions and express a semi-invariant TCR receptor. Unlike classic MHC-restricted T cells, NKT cells acquire their effector functions during development, and their activation following recognition of antigens presented in the context of CD1d molecules results in the rapid production of large amounts of effector cytokines [6].

The CD1 family of antigen presenting molecules are major histocompatibility complex (MHC) class I-like molecules and can be divided into three groups [7,8]. Group 1 is composed of CD1a, b, and c, Group 2-CD1d, and Group 3-CD1e, which is intracellular and plays a role in lipid loading. Group 1 CD1 molecule expression is limited to CD4 and CD8 double-positive thymocytes and professional antigen presenting cells, whereas Group 2 CD1d is more broadly expressed and is present on non-hematopoietic cells, including some cancer cells. CD1d-restricted NKT cells can be divided into subtypes based on T cell receptor (TCR) expression. Type I NKT cells express a rearranged invariant TCRα chain, Vα14Jα18 in mice and Vα24Jα18 in human that is associated with Vβ chains of limited diversity [9,10,11,12], are called invariant NKT (iNKT) cells, and are the focus of this review. As innate-like lymphocytes, iNKT cells differentiate into mature effector cells during thymic development. Therefore, type I NKT cells can be further divided into three subsets that mirror the T helper subtypes including NKT1, NKT2, and NKT17. These subsets are primarily identified by differences in the levels of the transcription factor, promyelocytic leukemia zinc finger (PLZF), following differentiation signals in the thymus including TCR engagement. α-Galactosylceramide (α-GalCer) is a potent activator of iNKT cells and has been well characterized [13,14,15]. Following their activation and increased expression of a large array of cell death-inducing effector molecules, including perforin, FAS ligand, and TRAIL, NKT cells, like other cytotoxic cells, such as NK and CTL, can induce cell death in tumor cells. Studies from several groups have demonstrated that treatment of mice with α-GalCer can lead to a significant reduction in tumor burden [16,17,18,19]; hence, clinical trials have been performed in order to evaluate the effectiveness of α-GalCer as a potential therapeutic immunomodulator of NKT cells [20,21,22,23,24,25]. In contrast to type I iNKT cells, type II NKT cells express diverse TCRs. Due to this diversity, type II NKT cells have been challenging to study and hence, significantly less is known about them. Type II NKT cells are CD1d-restricted, but are unresponsive to α-GalCer [26,27]. They have been investigated experimentally using CD1d-tetramers loaded with other lipid antigens, specifically phospholipids, sphingolipids, and glycerolipids. Given that type II NKT cells are thought to be present in higher numbers in humans, compared to type I NKT cells, gaining a better understanding of their regulation is critical. Herein, we review studies focused primarily on the modulation of human type I iNKT cells in specific types of solid tumors, then discuss barriers that block their therapeutic efficacy, and finally, suggest potential strategies that can be employed to effectively target NKT cells in cancer immunotherapeutic strategies.

## 2. Lung and Head and Neck Cancers

Lung cancer is the leading cause of cancer death worldwide, with an estimated 1.8 million deaths in 2020 [28]. Non-small cell lung cancer (NSCLC) accounts for approximately 85% of all lung cancer cases and mortality rates in NSCLC remain high due to poor detection methods and resistance to current treatment strategies. Thus, the development of new therapeutics is urgently needed. In a recent study, Dockry et al. examined NKT cell levels in the blood and lungs of NSCLC patients [29]. The investigators identified a significant reduction in NKT cells, as well as a reduction in CD1d in the lungs of NSCLC patients, compared to healthy lungs. Moreover, this reduction in CD1d was correlated with poor survival. The group also investigated the efficacy of DNA methyltransferase inhibitors (DNMTi) and histone deacetylase inhibitors (HDACi) in restoring CD1d expression [29]. The authors found that treatment with DNMTi and HDACi resulted in an increase in both CD1d mRNA and protein levels in NSCLC cell lines, thus sensitizing the cells to NKT cell mediated killing. These data are in good agreement with studies performed by our group and others showing that epigenetic modulators can induce CD1d expression in tumors and enhance NKT cell responses [30,31]. Motohashi and colleagues have conducted several clinical trials investigating the efficacy of NKT cell-based immunotherapy in NSCLC [24,32,33,34]. Overall, it was observed that NKT cell-based immunotherapy did not induce severe adverse events, the administration of α-GalCer-pulsed antigen presenting cells (APCs) demonstrated greater efficacy than the administration of ex vivo expanded iNKT cells, and an increase in IFN-γ producing cells in the periphery following treatment correlated with an increase in the median survival time (MST) (reviewed in [35]), see Table 1 for a summary of NKT cell-based clinical trials.

Similar to NSCLC, head and neck squamous cell carcinoma (HNSCC) remains a major health concern and it continues to rise in incidence. Currently, HNSCC is the sixth most common cancer worldwide, and is thought to be responsible for 450,000 deaths in 2018 [36]. Given the advances Motohashi and colleagues made in the development of NKT cell-based immunotherapy in NSCLC, they also developed strategies to effectively modulate NKT cells in HNSCC [35]. They examined IFN-γ production and the proliferative capacity of NKT cells in patients with stage IV HNSCC pre and post 50 Gy radiation therapy and found that radiation therapy did not negatively impact NKT cell number or function [37]. In other studies, it was found that nasal submucosa injection of α-GalCer-pulsed APCs led to higher numbers of NKT cells in the periphery, rather than injection of α-GalCer-pulsed APCs into the oral floor submucosa or intravenously [38,39]. Collectively the studies examining the clinical efficacy of NKT cell based immunotherapy in HNCC demonstrated the use of the intra-arterial infusion of activated Vα24 NKT cells in combination with submucosal injection of α-GalCer-pulsed APC resulted in antitumor immune responses [40,41].

## 3. Prostate Cancer

The intersection between immune cells, non-transformed stromal cells, and neovasculature helps to mitigate tumor cell progression. Dellabona and colleagues investigated iNKT cells, tumor associated macrophages (TAMs), and angiogenic factors in prostate cancer patient samples [42]. In this study, patients were stratified into low or high disease aggressiveness based on Gleason score, and it was found that patients with aggressive disease (Gleason score ≥ 8) had a significant accumulation of M2-like TAMs, fewer iNKT cells, and an enriched angiogenesis gene set compared to patients with less aggressive tumors [42]. The reduction in NKT cells in patients with advanced prostate cancer, myelodysplastic syndrome, and other solid tumors is associated with reduced proliferation and IFN-γ production [43]. In a study by Nowak et al., a murine prostate cancer model was used to evaluate if tumor cells inhibit the functions of NKT cells. The functional defects found in the murine prostate cancer model transgenic adenocarcinoma of the mouse prostate (TRAMP), were similar to those observed in human malignancies [44]. In brief, the authors demonstrated that co-culture of NKT cells with murine prostate cancer cells in the presence of α-GalCer resulted in modest activation due to cell-contact dependent inhibition of IL-12-mediated STAT4 phosphorylation in iNKT cells. However, the addition of α-GalCer and IL-12, restored IFN-γ production by NKT cells [44]. In addition, blockade of the Ly49 inhibitory NK receptor in the presence of α-GalCer pulsed-prostate cancer cells rescued IFN-γ production by iNKT cells. Future studies may employ strategies to either reduce tumor numbers or effects below this inhibitory threshold in order to boost NKT activation. The reversible nature of defects in the functionality and selectivity of NKT cells engenders the promising potential of their value as therapeutic targets in the realm of prostate cancer as well as other malignancies [6].

## 4. Brain Cancers and Neuroblastoma

Despite advances in the field, median survival for patients with brain malignancies remains poor [45]. Studies by Dhodapkar et al. [46], have shown that NKT cells are functionally and quantitatively similar in brain tumor patients and healthy controls, in contrast to other studies that have reported a decrease in NKT cells in advanced tumors [46,47]. CD1d was expressed on both primary glioma cells as well as endothelial cell glioma tissue sections, which suggest that modalities that selectively target NKT cells may have a significant impact in this malignancy. Neuroblastoma (NB) is a heterogenous tumor that is second most common in children [48]. It has been reported that in tumors in which the oncogene MYCN is amplified, there is a lack of NKT cells due to MYCN-mediated transcriptional repression of CCL2 [49]. Moreover, two-thirds of human NB cell lines secrete CCL2, which mediates transendothelial migration of iNKTs in vitro, therefore, iNKT cells migrate toward NB cells in a CCL2-dependent manner, preferentially infiltrating MYCN nonamplified tumors that express CCL2 [49]. A study has recently shown that NKT cells can serve as a prognostic factor in MYCN-non-amplified NB [50]. Additionally, NB cells selectively express high levels of the ganglioside, GD2. In studies utilizing T cells expressing a third generation GD2-CAR, the therapy was well tolerated; however, none of the patients with NB achieved an objective response [51]. Importantly, in a phase 1 clinical trial in children with relapsed or refractory NB, it was found anti-GD2 CAR-NKT cells expand and persist in these patients with NB, as well as infiltrate tumor sites and induce tumor regression in bone metastatic lesions [52]. This study paves way for expansion of this therapy to larger clinical trials and demonstrates the utility of NKT cell-based therapy for the treatment of NB patients.

## 5. Melanoma

Although a decrease in human NKT cells was observed in the peripheral blood of melanoma patients compared to healthy donors, patient NKT cells expanded similarly and mediated antitumor cytolytic activity against cell lines comparable to healthy donors [53]. These findings suggest that adoptive cell transfer of activated NKT cells or their modulation in vivo may have therapeutic efficacy in melanoma [53]. Ibarrando et al. characterized NKT cells in patients with metastatic melanoma pre and post treatment with an anti-CTLA4 mAb, tremelimumab, alone or in combination with MART-1 peptide-pulsed dendritic cells [54]. Both treatments resulted in the modulation of NKT cell number and subset distribution. Specifically, MART-1/DC responders experienced an increase in CD8+ iNKT cells and a concomitant decrease in CD4+ iNKT cells. Importantly, in patients treated with tremelimumab plus MART-1/DC, the responders (patients with a positive clinical outcome), had higher numbers of iNKT cells, compared to non-responders [54].

## 6. Colorectal Cancer and Renal Cell Carcinoma

Human colon-carcinoma infiltrating lymphocytes have been found to contain a small portion of presumably effective NKT cells, which has some value in prolonging survival rates—based on the degree of NKT infiltration [55,56]. In general, there is a positive correlation between the increased levels of NKT cells and a favorable prognosis. Compared to normal colorectal tissue, NKT cells were significantly higher in number in colorectal carcinomas [56]. Moreover, NKT cells are thought to play a vital role in the inhibition of lymph node metastasis of colorectal carcinomas. Coca et al. found that patients with extensive NKT cell infiltration into stage I and II tumors had a 100% 5-year relapse free survival rate [56]. However, there was no significant difference observed in the disease-free survival rate for those who had extensive NKT cell infiltration versus those with little/moderate NKT cell infiltration in stage I and II tumors [55]. In contrast, there was a significant difference in the 5-year relapse-free survival rate in patients with stage III colorectal carcinomas when extensive intratumoral NKT infiltration was compared with little/moderate infiltration. Remarkably, there was a relative ~ 4-fold difference in extensive NKT infiltration versus little/moderate infiltration. In conclusion, it is believed that in patients with colorectal carcinomas the extent of NKT infiltration is strongly correlated with the survival rate. This may be due to the ability of the intratumoral NKT cells to inhibit lymph node metastasis. These studies support the notion that NKT cell intratumoral infiltration can be a prognostic tool in colorectal carcinomas, particularly in stage III tumors. In a study investigating lymphocytes in colorectal cancer patients, a downregulation in the natural cytotoxicity receptors NKp44 and NKp46 was observed on circulating NKT-like cells in patients compared to healthy donors. The authors also found that a higher percentage of CD16^+^ NKT-like cells correlated with shorter disease-free survival in CRC patients [57]. Interestingly, Strober and colleagues identified a subset of type II CD1d-restricted, NKT cells that produced IL-13 and is associated with ulcerative colitis, but not Crohn’s disease [58].In fact, NKT cells play a key role in regulating intestinal inflammation [59,60]. Elegant studies by Blumberg and colleagues have shown that intestinal epithelial CD1d can impede intestinal inflammation through STAT3, IL-10, and HSP110-dependent mechanisms. In contrast, CD1d on bone marrow-derived cells can induce pathogenic NKT-cell activation in the intestines [61]. In murine models with colorectal metastases, treatment with α-GalCer significantly inhibited the progression of tumor growth and resulted in a high number of disease-free mice with acquired tumor-specific immunity [62]. Another study in which renal carcinoma cells (RCC) were evaluated for their expression of CD1d showed that CD1d was highly expressed in RCCs [63]. Taken together with the studies focused on CD1d expression by tumors, this suggests a role for agents that induce CD1d-mediated antigen presentation as means of enhancing NKT cell-mediated cancer immune surveillance in the tumor microenvironment (Figure 1).

## 7. Other Cancer Types

Studies from our group have demonstrated that ovarian cancer associated ascites contains suppressive factors that block NKT cell activation [65]. In addition, we have identified ganglioside GD3 and vascular endothelial growth factor (VEGF) as ovarian cancer associated immunosuppressive factors that inhibit CD1d-mediated NKT cell activation [66,67]. Of note, NKT cell activation was restored when VEGF was inhibited, subsequently leading to the reduction in expression of GD3. Work from another group has shown that ovarian cancers shed GD3 in exosomes [68], suggesting that we further investigate drugs, such as Avastin, that are known to impact tumor growth, because they may also increase anti-tumor immune responses. Similarly, a study by Fallarini et al., demonstrated that treatment of human osteosarcoma cells with chemotherapy (cisplatin, doxorubicin, and methotrexate) sensitized the cells to NKT cell-mediated cytotoxicity, in a CD1d-dependent manner [69]. Other pathways of NKT activation include inhibition of cyclooxygenase-2 (COX-2) as seen in a study by Klatka et al. [70] The study demonstrated that in patients with laryngeal cancer, COX-2 inhibition enhances NKT cells activation and proliferation further demonstrating its utility as an immunotherapy enhancing tool. In another study examining the role of NKT cell responses in the murine 4T1 breast cancer model, it was found that the combination of activated NKT cells with chemotherapeutic agents such as cyclophosphamide and gemcitabine, significantly enhanced survival in mice, and attenuated tumor growth following tumor rechallenge [71]. In another study by this group, treatment of tumor bearing mice with α-GalCer-loaded DCs decreased metastatic disease and enhanced survival that was associated with a reduction in MDSC, followed by the induction of antitumor responses by NKT cells and CD8+ T cells [72]. In addition, downregulation of CD1d in human breast cancer cells is correlated with increasing metastatic potential [73]. Another group has shown that CD1d downregulation by human papillomavirus (HPV) in infected cervical epithelial cells correlated with progression to cervical carcinoma [74]. Moreover, a recent study examined CD1d levels in anaplastic thyroid carcinomas and observed significantly higher levels of CD1d, compared to normal thyroid tissue [75], implicating a potential role for NKT cell-based therapy in this disease as well.

## 8. Potential Barriers to Successful Targeting of NKT Cells

While it is been well established that NKT cells can directly mediate tumor cell lysis, as well as rapidly produce cytokines following their activation, there are several factors that impede their effectiveness in cancer immunotherapy. As stated above, NKT cells are numerically reduced and functionally impaired in many cancer types. Such circumstances may include the loss of CD1d expression, due to its down regulation or the loss of β2-microglobulin, or a lack of activating antigens. In addition, immunosuppressive cells and factors in the tumor microenvironment are obstacles to successful immunotherapy in patients. NKT cells communicate and interact with many of these suppressive cell types, such as myeloid derived suppressor cells (MDSCs) and Tregs. A subset of NKT cells, type II NKT cells, have been shown to directly inhibit anti-tumor responses.

Several studies have examined the crosstalk between iNKT and MDSCs. MDSCs are a population of immature myeloid cells that accumulate in the blood and tumor microenvironment in cancer patients [76]. MDSCs typically express CD11b, CD33, and low levels of HLA-DR in humans or CD11b, Gr1, and CD124 in mice [77]. MDSCs promote an immunosuppressive environment by secreting arginase I, nitrogen oxide, reactive oxygen species, and TGF-β, as well as through direct cell-contact mediated mechanisms. It has been shown that α-GalCer-pulsed MDSCs can be converted into immunogenic antigen presenting cells that induce immune responses and lead to increased survival in Her-2/CT26 tumor-bearing mice [78]. In addition, Dr. Metelitsa’s group investigated tumor associated macrophages (TAMs) in neuroblastoma and found that while CD1d+TAMs can promote tumor growth, TAMs can also present neuroblastoma antigens that can lead to their recognition and killing by NKT cells [79].

Tregs are another type of immunosuppressive cell expanded in cancer and involved in immunological self-tolerance. Tregs express the transcription factor FoxP3, a transcriptional regulator associated with suppressive activity, and develop in the thymus or periphery when activated by TGF-β. Monteiro et al. discovered a unique population of murine Foxp3+ NKT cells that display typical Treg surface molecules, are activated by TGF- β and suppress T cell proliferation via an IL-10-dependent mechanism [80]. The group induced autoimmune encephalomyelitis (EAE) in mice and then treated with α-GalCer, which protected mice from the disease [80]. Furthermore, it was found TGF-β promotes Foxp3 upregulation in both NKT cells and CD4+ T cells, converting them into suppressive Foxp3+ cells in both instances. Another study using human peripheral blood mononuclear cells and cord blood mononuclear cells showed that NKT cells can express FoxP3 after exposure to TGF-β, but suppressive activity was only conferred after treatment with rapamycin, an mTOR pathway inhibitor that promotes expansion of Tregs [81]. It was also found that FoxP3+ iNKT cells can suppress the proliferation of human CD4+ cells [81]. Further research is necessary to elucidate the role of FoxP3+ NKT cells in tumor immunosurveillance and immunotherapy. In a murine study investigating the role of NKT cells in tumor immunosurveillance, it was found that type I NKT cell deficient (Jα18^−/−^) and WT mice developed similar numbers of lung metastases, whereas CD1KO mice developed significantly fewer metastases [82]. When the group depleted CD4+ CD25+ Tregs in Jα18^−/−^ mice, no effect was observed on the development of pulmonary metastases [82]. Their results indicate that in the absence of both type I NKT cells and Tregs, type II NKT cells are sufficient to suppress immunosurveillance.

### Type II NKT Cells

Type I and type II NKT cells are reported to have counter-regulatory roles in tumor immune surveillance. Several studies have found type II NKT cells have a suppressive phenotype, whereas type I NKT cells bolster protective anti-tumor activity (reviewed in [83]). In the tumor microenvironment, type II NKT cell-induced tumor suppression can be mediated by IL-13 secretion resulting in the activation of TGF-β-secreting MDSCs that inhibit tumor-specific CD8+ T cells or type I NKT cells [83]. One known type II NKT cell agonist is sulfatide, which is a naturally occurring self-glycolipid in many tissues [84,85]. Another type-II NKT cell agonist, lysophosphatidylcholine (LPC) was found to be enriched in myeloma patients’ plasma samples. The LPC-reactive type II cells showed a Th2-type (IL-13) response, implying an immunosuppressive phenotype [86]. Ambrosino et al. presented the first evidence for cross-regulation between type I and type II NKT cells in a murine model of CT26 and 15–12RM tumors [87]. In both WT mice and Jα18^−/−^ mice, treatment with sulfatide increased the number of lung nodules, while treatment did not affect tumor growth in CD1KO [87]. This provided further evidence that type II NKT cells reduce anti-tumor immunity. In contrast, in vivo activation of type I NKT cells with α-GalCer completely protected the mice from tumor growth, suggesting type I NKT cells take on an opposing role to type II NKT cells [87].

## 9. Future Directions

### 9.1. Chimeric Antigen Receptor (CAR)-NKT

Chimeric antigen receptor (CAR) based-immunotherapy has had limited clinical efficacy in solid tumors, partially due to the immunosuppressive tumor microenvironment. Recent interest in using an NKT cell carrier rather than the standard T cell has gained traction, since CD1d is relatively monomorphic, has limited expression, and a reduced risk of graft-versus-host disease (GVHD) [88,89]. Compelling preclinical studies from Dr. Yang’s group has shown that hematopoietic stem cell engineered iNKT (HSC-iNKT) cell-based therapy can be highly effective for the treatment of hematologic malignancies using a human multiple myeloma xenograft NSG mouse model [88]. Metelitsa and colleagues have conducted extensive studies investigating the efficacy of CAR-NKT cells. Heczey et al. found CAR.GD2 NKT cells had potent antitumor activity against GD2 positive tumor cells in a metastatic NB murine model [90]. Additionally, unlike T cells, the NKT platform did not induce graft-versus-host disease (GVHD) [90]. In addition to tumor regression, CAR.GD2 NKT cells also eliminated immunosuppressive CD1d-positive M2 macrophages in vitro [90]. Rotolo et al. optimized CAR-NKT cell production and showed an anti-tumor effect of CAR19-iNKT over CAR-T cells in terms of both tumor-free and overall survival in an in vivo mouse model of CD1d+CD19+ B lineage cancer. The CAR-NKT platform did not result in GVHD [91]. They also found that only a single dose of CAR19-iNKT was required for drastic tumor clearance [91]. These studies indicate NKT cells have greater potential in CAR immunotherapy for solid tumors compared to conventional T cells.

### 9.2. Checkpoint Inhibitor Therapy

Programmed death (PD)-1 expression is found in activated T cells, B cells, NK cells, and NKT cell and serves as a regulator of immunological self-tolerance. Tumors upregulate PDL-1 to escape tumor immune surveillance. Due to their innate-like nature, NKT cell-mediated tumor immunotherapy has been limited because their stimulation can be followed by long term anergy. One form of immunotherapy involves blocking PD-1/PDL-1 interactions to minimize this anergic response of NKT cells following stimulation. Parekh et al. sought to overcome α-GalCer induced NKT anergy in mice by using antibodies against PD-1 [92]. They found PD-1/PD-L1 blockade at the time of α-GalCer treatment prevented induction of NKT cell anergy and maintained anti-tumor effects. However, after the anergic phenotype had been established, PD-1/PDL-1 blockade did not affect NKT responsiveness [92]. Interestingly, when NKT cells were treated with other bacterial or sulfatide agonists, blockade of the PD-1/PD-L1 pathway failed to prevent NKT cell anergy, suggesting additional mechanisms at play [92]. Another study probed the interaction between PD-1 on iNKT cells and PD-L1 on antigen-presenting cells (APCs) [93]. Kamata et al. found PD-L1 blockade at the time of stimulation resulted in increased release of helper Th1 cytokines from NKT cells, leading to the activation of NK cells [93]. Using a human lung cancer cell line NCI-H460, they found NKT cells cultured with α-GalCer and APCs treated with anti-PDL1 Ab had direct tumor cytotoxicity [93].

### 9.3. Bifunctional Molecules

Bi-specific T cell engagers (BiTEs) are genetically engineered recombinant proteins that have been shown to activate cytotoxic T cells through the CD3 complex, while also targeting activated T cells to tumor cells via a tumor-specific domain. Horn et al. developed a novel BiTE construct consisting of anti-CD3 mAb linked to an anti-PD-L1 mAb (CD3xPDL1) to bridge activated T cells to PDL1-expressing tumors [94]. It was discovered in vitro that the CD3xPDL1 BiTE activates T cells as well as NKT cells from healthy donors and cancer patients, and that the activated cells were cytotoxic for multiple types of PD-L1 expressing human tumors [94]. In vivo studies demonstrated that the CD3xPDL1 BiTE significantly prolonged the mean survival time of NSG mice carrying spontaneously metastatic human melanoma tumors [94]. Another surprising result was that BiTE-treated mice had fewer mouse MDSC and a much lower ratio of mouse MDSC to human T cells [94]. Another group created a novel bifunctional molecule wherein soluble CD1d was fused to an anti-HER2 antibody. Systemic injections of this molecule demonstrated potent inhibition of HER2-expressing lung tumors and established tumors [95]. Notably, even when treatment was delayed after the injection of the HER2- expressing B16 melanoma cells, there was still 60% less metastatic invasion in the α-GalCer/sCD1d–anti-HER2 treated mice than in controls or α-GalCer-only treated animals. Similarly, established tumors treated with the α-GalCer/sCD1d–anti-HER2 fusion protein seven days after the graft, were on average 60% smaller than in controls or α-GalCer-treated animals [95]. In good agreement, Corgnac et al. demonstrated consistent NKT-cell-mediated antitumor effects through repeated administration of tumor-targeted recombinant sCD1d-antitumor scFv fusion proteins specific for either HER2 or CEA, loaded with α-GalCer [96]. One group used an innovative approach to generate CD1d-BiTEs that could be used to repeatedly stimulate NKT cells without inducing anergy [97]. These BiTEs contained photoreactive analogues of α-GalCer, and using mouse models of colorectal cancer, it was found these constructs activated NKT cell effector functions, as well as induced epitope spreading for tumor-specific CD8+ cytolytic T-cell responses [97].

In summary, the field of cancer immunotherapy is rapidly evolving. Given the limited toxicity that has been observed with NKT cell-based therapeutics and the ability of NKT cells to directly kill tumor cells as well as effectively modulating other immune cells, several groups are exploring combinatorial strategies using checkpoint inhibitors and bispecific T cell engagers [97], developing unique platforms using nanotechnology [98], as well as harnessing the expansion potential of stem cells [88] to target this potent population of cells (Figure 2). Moreover, preclinical studies are investigating the efficacy of NKT cell-based therapy in combination with oncolytic therapy [99]. It is a great time to work in the field and with a greater understanding of their metabolic functions and the impact of co-signaling on subset differentiation and effector function, NKT cell-based treatments can reach their true potential in the field of cancer immunotherapy.

## Figures and Tables

**Figure 1 cells-10-01329-f001:**
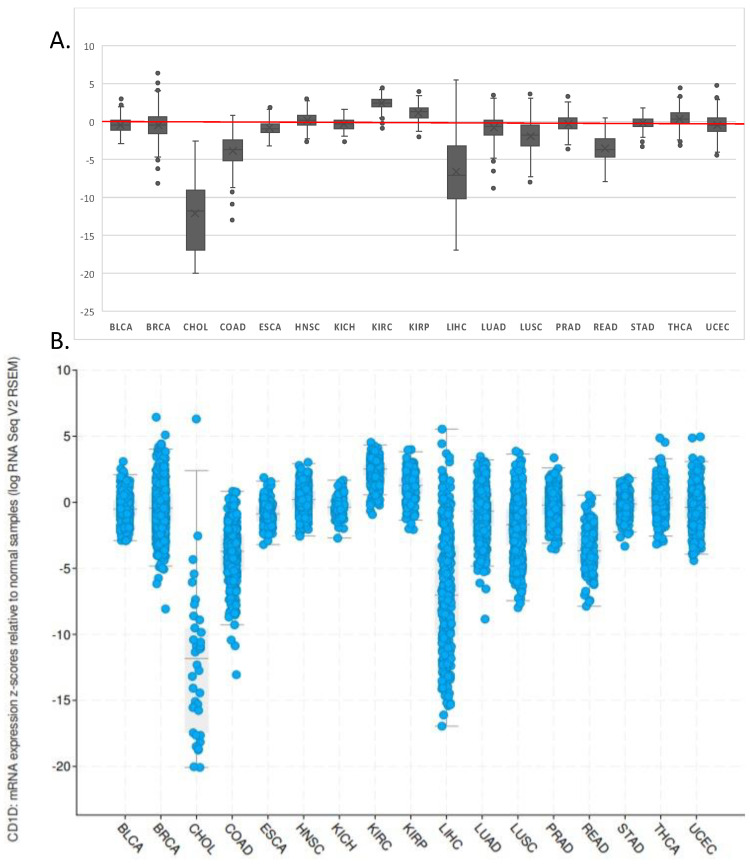
CD1D expression in different cancer types compared to corresponding healthy tissues. NKT cells can recognize and mediate cytolysis, thus CD1d-expression on malignant cells is an area of active investigation. It is thought that higher expression of CD1d leads to in higher tumor cell killing. Thus, several tumors have been shown to downregulate CD1d expression, perhaps in order to escape NKT cell-mediated immunosurveillance, or these data reflect a positive selection pressure for CD1d-low tumor cells. cBioPortal was used to investigate CD1D expression in different cancer types available in the TCGA database [64]. Data shown represents CD1d mRNA expression z scores relative to normal samples. (**A**) Median levels of CD1D is shown and 0 (no change) is indicated by a red line. (**B**) Each patient sample is indicated by a blue circle. BLCA = bladder urothelial carcinoma; BRCA = breast invasive carcinoma; CHOL = cholangiocarcinoma; COAD = colon adenocarcinoma; ESCA = Esophageal carcinoma; HNSC = Head and Neck squamous cell carcinoma; KICH = Kidney chromophobe; KIRC = Kidney clear cell carcinoma; KIRP = Kidney renal papillary cell carcinoma; LIHC = Liver hepatocellular carcinoma; LUAD = lung adenocarcinoma; LUSC = lung squamous cell carcinoma; PRAD = Prostate adenocarcinoma; READ = Rectum adenocarcinoma; STAD = Stomach adenocarcinoma; HCA = Thyroid carcinoma; UCEC = Uterine Corpus Endometrial Carcinoma.

**Figure 2 cells-10-01329-f002:**
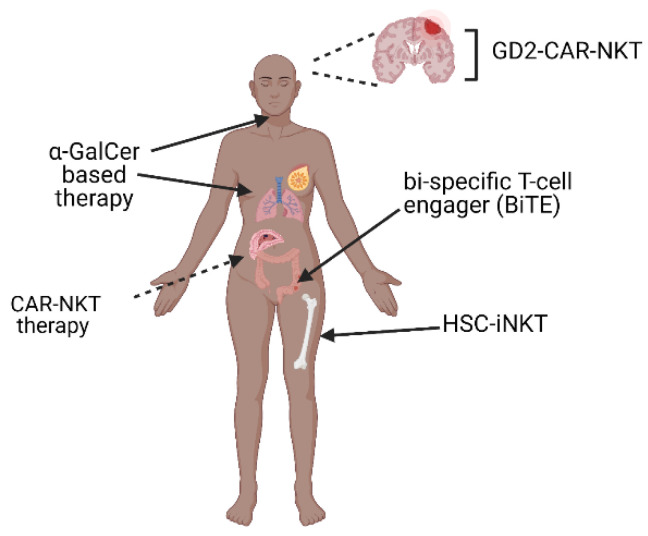
Targeting invariant natural killer T (iNKT) cells in solid tumors. Many strategies are being employed to harness the power of NKT cells in cancer immunotherapy. Treatment with α-GalCer-pulsed APC has been shown to result in antitumor immune responses, particularly in lung and head and neck cancers. Cell based therapeutics such as CAR-NKT and HSC-iNKT cells have shown efficacy in preclinical models. In addition, there are several reports investigating the direct modulation of NKT cells using BiTEs and α-GalCer-based therapy. Taken together, these studies demonstrate the feasibility of targeting NKT cells for cancer immunotherapy and present innovative strategies that can be employed to increase our understanding of this important population of T cells.

**Table 1 cells-10-01329-t001:** Clinical trials using NKT cell-based therapy.

**Trial ID**	**Reference**	**Phase**	**Cancer Type**	**Number of Patients**	**Date**	**Country**	**Treatment**
**Adoptive transfer of in vitro expanded autologous NKT cells**							
	Motohashi et al	I	Non-small cell Lung Cancer	6	2004–2006	Japan	Infusion of iNKT
NCT00631072	Exley et al	I	Melanoma	9	2008–2017	United States	Infusion of iNKT + GM-CSF
NCT00909558	Clinicaltrials.gov (accessed on 26 May 2021)	I	Breast Cancer Glioma Hepatocellular Cancer Squamous Cell Lung Cancer Pancreatic Cancer Colon Cancer	24 (estimated)	2009–	United States	Infusion of NK and NKT
NCT01801852	Clinicaltrials.gov (accessed on 26 May 2021)	I	Breast Cancer Glioma Hepatocellular Cancer Squamous Cell Lung Cancer Pancreatic cancer Colon Cancer Prostate Cancer	300 (estimated)	2013–	China	Infusion of NKT
NCT02619058	Clinicaltrials.gov (accessed on 26 May 2021)	I	Melanoma	20 (estimated)	2015–	China	Infusion of NKT
NCT02562963	Clinicaltrials.gov (accessed on 26 May 2021)	I–II	Advanced Solid Tumor	120 (estimated)	2015–	China	Infusion of NKT
NCT03093688	Clinicaltrials.gov (accessed on 26 May 2021)	I	Advanced Solid Tumor	40 (estimated)	2017–	China	Infusion of iNKT and CD8+ T cells
NCT03198923	Clinicaltrials.gov (accessed on 26 May 2021)	I	Non-small cell Lung Cancer	30 (estimated)	2017	China	Infusion of NK and NKT
NCT04011033	Clinicaltrials.gov (accessed on 26 May 2021)	II–III	Hepatocellular Carcinoma	144 (estimated)	2019–	China	Infusion of iNKT combined with TACE procedure
**Adoptive transfer of in vitro expanded allogenic NKT cells**							
NCT04754100	Clinicaltrials.gov (accessed on 26 May 2021)	I	Multiple Myeloma	30 (estimated)	2021–	United States	Infusion of agenT-797 iNKT therapy
**In vitro generated antigen presenting cells loaded with αGalCer**							
UMIN000007321	Ishikawa et al	I	Non-small cell Lung Cancer	11	2001–2002	Japan	Infusion of αGalCer-pulsed dendritic cells
	Motohashi et al	I	Non-small cell Lung Caner	23	2003–2004	Japan	Infusion of αGalCer-pulsed IL2/GM-CSF cultured PBMCs
NCT00698776	Richter et al	I	Myeloma	6	2009–2011	United States	Infusion of KRN7000-pulsed dendritic cells + Lenalidomide
**Type I NKT chimeric antigen receptor (CAR)**							
NCT03294954	Heczey et al	I	Neuroblastoma	24	2018–	United States	Autologous GD2-CAR NKT cells expressing IL-15
NCT03774654	Clinicaltrials.gov (accessed on 26 May 2021)	I	B cell Malignancies	48 (estimated)	2020–	United States	Allogenic CD19-CAR NKT cells expressing IL-15
NCT04814004	Clinicaltrials.gov (accessed on 26 May 2021)	I	Acute Lymphoblastic Leukemia B-cell Lymphoma Chronic Lymphocytic Leukemia	20 (estimated)	2021–	China	Autologous CD19-CAR iNKT cells expressing IL-15
**Other**							
NCT04751786	Clinicaltrials.gov (accessed on 26 May 2021)	I	Advanced Solid Tumor (NY-ESO-1 positive)	15 (estimated)	2021–	Netherlands	Administration of PRECIOUS-01, an iNKT cell activator threitolcermaide-6 and NY-ESO-1 encapsulated in a nanoparticle

## Data Availability

Publicly available datasets were analyzed in this study. These data can be found here: https://www.cbioportal.org/ (accessed on 26 May 2021).

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
