# Peer review of "Targeting Natural Killer T Cells in Solid Malignancies"

_cells, 2021, doi:10.3390/cells10061329_

Round 1
Reviewer 1 Report
In this manuscript by Ingram et al, the authors review the role of different types of NKT cells in anti-tumor immunity, as well as strategies to use the therapeutic potential of these cells in the treatment of solid tumors, either directly or in combination with other forms of cancer immunotherapy.
This is a well-written review describing the various studies that have addressed mechanisms of NKT cell-mediated anti-tumor activity in animal models, the association between the frequencies of these cells in cancer patients and disease prognosis in different cancer types, in addition to a description of the feasibility and the hurdles to be overcome before the successful use of NKT cell-based therapeutics.
1 My only critism is the description of type I and type II NKT cells in the introduction and the text. It is mentioned in the introduction that CD1d restricted NKT cells can be divided into subtypes based on T cell receptor (TCR) expression and type I NKT cells,…, called invariant chain NKT (iNKT) cells are the focus of their review. However, the authors refer to NKT, and not iNKT, cells throughout the text. Furthermore, they subsequently detail type I but not type II NKT cells. The latter are mentioned for the first time section 8.1. Although the latter section should of course be maintained it is recommended to include a description of the type II NKT cells in the introduction, as part of the description of the various subtypes (see also under 3).
2 It is of note that many of the references refer to studies carried out more than 10 years ago or longer and that the review contains relatively few recent references which seems somehow points to a lack of progress in the field with respect to feasibilty and therapeutic benefits of these cells in immune therapy for human cancer patients. Even the CAR-iNKT section contains only two studies dealing with progress in this area given the interest that dates back seven years ago.
References 51 and 69 cannot be considered recent. This also pertains to “recent interest” (Line 280) going back 2014).
3 It is stated in line 52 that NKT cells numbers are reduced in multiple cancer types. This is certainly true for type I cells, but what about type II NKT cells taking into accoun the observation that the type-II NKT cell agonist, lysophosphatidylcholine (LPC) 267 is enriched in plasma of myeloma patients. It would be more precise to refer to type I NKT numbers, hence the remark to include the type II cells in the introduction.
4 It is somewhat intriguing that the immune suppression mediated by IL-13 via the activation of TGF-β-secreting MDSC in only exerted by type II NKT cells. Is this immunosuppressive capacity linked with the difference in antigen recognition between type II and type I NKT cells, given that the latter also produce IL-13 (at least the NKT2 cells), but apparently do not exert immuno suppressive effects ?
Some corrections in the text:
Line 37 ..both the innate..
Line 39 ..transplantation..
Line 40 CD1-restricted ..
Line 41-43 This phrase is somewhat confusing : One suggestion is : Type I NKT cells express a rearranged invariant TCRα chain, Vα14Jα18 in mice and Vα24Jα18 in human, that is associated with different Vβ chains of limited diversity.
Line 39 ..transplantation..
Line 50-52 Following their activation and increased expression of a large array of cell death-inducing effector molecules, including perforin, FAS ligand, and TRAIL, NKT cells, like other cytotoxic cells such as NK cells and CTL, can induce cell death in tumor cells.
Line 53 .. that treatment..
Line 203 .. NKT cell activation..
Line 214 .. directly mediate..
Line 230 .. that α-GalCer-pulsed MDSCs can..
Line 233 and found that CD1d+ while TAMs can promote ß?
Line 269 …IL-3 is likely a typo and should read IL-13.
Line 333 ..repeated administration..
Author Response
Thank you for your careful review of our manuscript. We appreciate your comments as they have helped to provide clarity and significantly strengthen our article.
Response to Reviewer 1 Comments
1 My only critism is the description of type I and type II NKT cells in the introduction and the text. It is mentioned in the introduction that CD1d restricted NKT cells can be divided into subtypes based on T cell receptor (TCR) expression and type I NKT cells,…, called invariant chain NKT (iNKT) cells are the focus of their review. However, the authors refer to NKT, and not iNKT, cells throughout the text. Furthermore, they subsequently detail type I but not type II NKT cells. The latter are mentioned for the first time section 8.1. Although the latter section should of course be maintained it is recommended to include a description of the type II NKT cells in the introduction, as part of the description of the various subtypes (see also under 3).
Thanks for bringing this to our attention. We have added a paragraph to the intro: “In contrast to type I iNKT cells, type II NKT cells express diverse TCRs. Due to this diversity, type II NKT cells have been challenging to study and hence, significantly much less is known about them. Type II NKT cells are CD1d-restricted, but are unresponsive to α-GalCer. They have been investigated experimentally using CD1d-tetramers loaded with other lipid anti-gens, specifically phospholipids, sphingolipids, and glycerolipids. Given that type II NKT cells are thought to be present in higher numbers in humans, compared to type I NKT cells, gaining a better understanding of their regulation is critical.”
2 It is of note that many of the references refer to studies carried out more than 10 years ago or longer and that the review contains relatively few recent references which seems somehow points to a lack of progress in the field with respect to feasibilty and therapeutic benefits of these cells in immune therapy for human cancer patients. Even the CAR-iNKT section contains only two studies dealing with progress in this area given the interest that dates back seven years ago.
Current references have been added to the text to highlight the progress that has been made in the field. In addition, Table 1 shows the NKT cell-based clinical trials further demonstrating the interest in NKT cell based therapeutics.
References 51 and 69 cannot be considered recent. This also pertains to “recent interest” (Line 280) going back 2014).
Thank you for bringing this to our attention. We have removed the word “recent” from line 250 (source 69) and line 151 source (50). In line 281 we have removed the word “recent” as well.
3 It is stated in line 52 that NKT cells numbers are reduced in multiple cancer types. This is certainly true for type I cells, but what about type II NKT cells taking into account the observation that the type-II NKT cell agonist, lysophosphatidylcholine (LPC) 267 is enriched in plasma of myeloma patients. It would be more precise to refer to type I NKT numbers, hence the remark to include the type II cells in the introduction.
Thank you, we have modified the text to accurately reflect our knowledge of the type I NKT cell population and added background on type II NKT cells.
4 It is somewhat intriguing that the immune suppression mediated by IL-13 via the activation of TGF-β-secreting MDSC in only exerted by type II NKT cells. Is this immunosuppressive capacity linked with the difference in antigen recognition between type II and type I NKT cells, given that the latter also produce IL-13 (at least the NKT2 cells), but apparently do not exert immuno suppressive effects ?
This is intriguing, however, given that both subtypes of NKT cells can produce immunosuppressive cytokines we do not know how much antigen recognition contributes to suppression mediated by type II NKT cells compared to the suppressive activities of type I iNKT cells.
Some corrections in the text:
Line 37 ..both the innate..
Line 39 ..transplantation..
Thank you for pointing this out. The word “transplant” has been changed to “transplantation” on line 34.
Line 40 CD1-restricted ..
Thank you for your suggestion. Respectfully, we did not make the change since the focus of our review is on CD1d restricted NKT cells. To aid in clarity, however, we added lines 41-46 in order to describe the broader CD1 family of antigen presenting molecules in detail.
Line 41-43 This phrase is somewhat confusing : One suggestion is : Type I NKT cells express a rearranged invariant TCRα chain, Vα14Jα18 in mice and Vα24Jα18 in human, that is associated with different Vβ chains of limited diversity.
We agree that this sentence lacks clarity and have updated lines 42-43 to read, “Type I NKT cells express a rearranged invariant TCRα chain, Vα14Jα18 in mice and Vα24Jα18 in human, that is associated with Vβ chains of limited diversity [7-10], called invariant NKT (iNKT) cells, and are the focus of this review”.
Line 39 ..transplantation..
Thank you for pointing this out. The word “transplant” has been changed to “transplantation” on line 34
Line 50-52 Following their activation and increased expression of a large array of cell death-inducing effector molecules, including perforin, FAS ligand, and TRAIL, NKT cells, like other cytotoxic cells such as NK cells and CTL, can induce cell death in tumor cells.
Thank you for this correction, as our wording failed to specifically identify NKT cells in our discussion of their similarities to other cytotoxic cells. We have added these changes to lines 50-52.
Line 53 .. that treatment..
Thank you for pointing out this grammatical error. We have removed the comma after the word “that” in line 53.
Line 203 .. NKT cell activation..
Line 214 .. directly mediate..
Thank you, the grammatical errors have been corrected.
Line 230 .. that α-GalCer-pulsed MDSCs can..
Thank you, the grammatical error has been corrected on line 236.
Line 233 and found that CD1d+ while TAMs can promote ß?
Thank you for pointing this out, we changed line 246 so that the intended meaning is clear.
Line 269 …IL-3 is likely a typo and should read IL-13.
Thank you for finding our error, “IL3” has been corrected to “IL-13” on line 279.
Line 333 ..repeated administration..
Thank you, the grammatical error has been corrected (now on line 337).
Reviewer 2 Report
The author provides a concise summary of the pathological involvement of NKT cells and their treatment for solid tumors. Overall, this manuscript is very well organized and concise, making it easy to understand for readers and also provides an overview of the relevance of modern NKT therapies to solid tumors.
Major points 
1.The data in summary on solid cancer treatment is somewhat outdated. Especially for colorectal cancer, the latest findings are not included.
2.There are many references to CD1d expression in particular. It is desirable to create a table on CD1d expression for normal cells and tumor cells.
Minor ponts
Is the position of "CD1+" in line 233 correct?
Author Response
Thank you for taking the time to critique our review. We appreciate your efforts and believe that our manuscript is stronger and much more comprehensive.
Response to Reviewer 2
Major points 
- The data in summary on solid cancer treatment is somewhat outdated. Especially for colorectal cancer, the latest findings are not included.
Thank you for bringing this to our attention, this addition to the text was made:
In a study investigating lymphocytes in colorectal cancer patients, a downregulation in the natural cytotoxicity receptors NKp44 and NKp46 was observed on circulating NKT-like cells in patients compared to healthy donors. The authors also found that a higher percentage of CD16+ NKT-like cells correlated with shorter disease-free survival in CRC patients [61]. Interestingly, Strober and colleagues identified a subset of type II CD1d-restricted, NKT cells that produced IL-13 and is associated with ulcerative colitis, but not Crohn’s disease [62].In fact, NKT cells play a key role in regulating intestinal inflammation [63, 64]. Elegant studies by Blumberg and colleagues have shown that intestinal epithelial CD1d can impede intestinal inflammation through STAT3, IL-10 and HSP110-dependent mechanisms. In contrast, CD1d on bone marrow-derived cells can induce pathogenic NKT-cell activation in the intestines [65].
- There are many references to CD1d expression in particular. It is desirable to create a table on CD1d expression for normal cells and tumor cells.
We agree with your suggestion and have incorporated Figure 1, which uses data gathered from CBioPortal to display CD1d mRNA expression in different cancer types compared to corresponding healthy tissues.
Minor points
- Is the position of "CD1+" in line 233 correct?
Thank you for the suggestion, we have made the appropriate change to line 282 and hope that the sentence is now clear.
Reviewer 3 Report
Reviewer Comments
Zewde Ingram et al. in this review they reported studies investigating the therapeutic use of NKT cells in different solid tumors. This manuscript is very interesting and could be very useful for many readers involved in the studies of NKT cells and their therapeutic potential in the context of solid tumors for which great efforts are still needed.
The review is structured with an introduction, five paragraphs describing the studies on NKT cells in solid tumors and a paragraph, organized into subsections, devoted to the strategies adopted to overcome tumor barriers to successfully target NKT cells. The manuscript needs to be revised in many points. In details below I indicate some suggestions aimed at improving this manuscript:
- In Introduction the authors reported a-GalCer as a potent activator of NKT cells. Since a-GalCer is described as the specific ligand for NKT cells in many papers, having a crucial role on NKT function, details on the molecular mechanisms leading to such activation, as well as further references, should be added here;
- In introduction, page 2, line 52 delete “.” after “in tumor cells”;
- In the paragraph “Lung and Head and Neck Cancers”, page 2, lines 74-76, the sentences should be revised (“in NSCLC cell lines, sensitized……”, better “thus sensitizing…);
- In the same paragraph, page 2, lines 76-83, the authors reported studies on lymphoma, breast cancer cells, cervical epithelial cells and thyroid carcinomas that could be moved to the paragraph entitled “Other Cancer Types” where they are still included the types of cancer associated with these;
- In paragraph “Prostate Cancer”, page 3, line 111, reference 40 appears to be inappropriate as it describes a clinical trial in patients with melanoma. Otherwise, the sentence must be rewritten; in line 113, the authors reported that reduction of NKT cells in cancer patients is associated with reduced proliferation and IFN-gamma production, with reference 40 appearing to be non-exhaustive and must be added to reference 46;
- In the same paragraph, line 118-120, the authors reported that NKT cells are partially activated by tumor cells while INF-gamma production is inhibited….the authors should describe how or in what terms NKT cells are activated and put a suitable reference;
- In paragraph “Brain Cancer”, page 3, line 130, the reference 44 in the group 44-46 should be deleted since in this sentence the authors referred to the decrease of NKT cells in advanced solid tumors;
- In same paragraph, the authors incorrectly reported studies on Neuroblastoma which is not a brain cancer but is an extracranial tumor. So the paragraph tittle should be correct as Brain Cancers and Neuroblastoma;
- In the paragraph “Melanoma”, the first sentence in lines 150-154 is too long, unclear and needs a reference. Authors should rewrite these concepts. In line 158 “subset distribution” it is not clear which cell type this refers to;
- In paragraph “Colorectal cancer and Renal cell carcinoma” line 173 correct “significance difference” with” significant difference”; line 174 correct “compared against” with “ compare with”;
- In the same paragraph, line 178, delete “in the patient population” as it is redundant.
- In line 179, authors refer to NK cells (?);
- In line 182 “was attributed” referred to a-GalCer treatment is unclear and should be appropriately rewrite;
- In lines 183-188 the sentences must be rewritten with care as they unclear and contain grammatical errors;
- In the paragraph “Other Cancer Types”, page 5, in line 211, add the explanation of the acronyms for MDSC by moving it from the line 222;
- In line 214 correct “can directly mediated” with “can directly mediate”;
- In line 215 correct “;” with “,”;
- In lines 232-234 something seems to be missing after “found that CD1d+”…. carefully review the whole sentence;
- In line 273 review the sentence “this was provided further evidence”
- On page 12, line 341, the authors should add a paragraphs of conclusions to summarize the meanings of the review, the description of therapeutic strategies referred to Figure 1 and add a Table with all current clinical trials on the use of NKT cells for solid tumors, by visiting clinicaltrial.gov website.
Author Response
Response to Reviewer 3
Thank you for your thorough review of our manuscript. We are very appreciative of your criticism and believe that your critiques have helped to greatly improve our article. Please find below our point-by-point response in bold font.
The review is structured with an introduction, five paragraphs describing the studies on NKT cells in solid tumors and a paragraph, organized into subsections, devoted to the strategies adopted to overcome tumor barriers to successfully target NKT cells. The manuscript needs to be revised in many points. In details below I indicate some suggestions aimed at improving this manuscript:
- In Introduction the authors reported a-GalCer as a potent activator of NKT cells. Since a-GalCer is described as the specific ligand for NKT cells in many papers, having a crucial role on NKT function, details on the molecular mechanisms leading to such activation, as well as further references, should be added here;
Two additional references about a-GalCer have been added:
14. M. Morita et al., "Structure-activity relationship of alpha-galactosylceramides against B16-bearing mice," (in eng), J Med Chem, vol. 38, no. 12, pp. 2176-87, Jun 9 1995, doi: 10.1021/jm00012a018.
15. J. Rossjohn, D. G. Pellicci, O. Patel, L. Gapin, and D. I. Godfrey, "Recognition of CD1d-restricted antigens by natural killer T cells," (in eng), Nat Rev Immunol, vol. 12, no. 12, pp. 845-57, Dec 2012, doi: 10.1038/nri3328.
- In introduction, page 2, line 52 delete “.” after “in tumor cells”;
Thank you for pointing this out, the grammatical error has been corrected.
- In the paragraph “Lung and Head and Neck Cancers”, page 2, lines 74-76, the sentences should be revised (“in NSCLC cell lines, sensitized……”, better “thus sensitizing…);
Thank you for the suggestion, we have corrected the sentence from lines 77-79 to read, “The authors found that treatment with DNMTi and HDACs resulted in an increase in both CD1d mRNA and protein levels in NSCLC cell lines, thus sensitizing the cells to NKT cell mediated killing”.
- ;
- In paragraph “Prostate Cancer”, page 3, line 111, reference 40 appears to be inappropriate as it describes a clinical trial in patients with melanoma. Otherwise, the sentence must be rewritten; in line 113, the authors reported that reduction of NKT cells in cancer patients is associated with reduced proliferation and IFN-gamma production, with reference 40 appearing to be non-exhaustive and must be added to reference 46;
We apologize for this error in formatting the references and thank you for bringing it to our attention. We have added additional references as well.
- In the same paragraph, line 118-120, the authors reported that NKT cells are partially activated by tumor cells while INF-gamma production is inhibited….the authors should describe how or in what terms NKT cells are activated and put a suitable reference;
Thank you, more experimental details have been added. Specifically, this statement was added: “In brief, the authors demonstrated that co-culture of NKT cells with murine prostate cancer cells in the presence of α-GalCer resulted in modest activation due to cell-contact dependent inhibition of IL-12-mediated STAT4 phosphorylation in iNKT cells. However, the addition of α-GalCer and IL-12, restored IFN-γ production by NKT cells [44]. In addition, blockade of the Ly49 inhibitory NK receptor in the presence of α-GalCer pulsed-prostate cancer cells rescued IFN-γ production by iNKT cells. Future studies may employ strategies to either reduce tumor numbers or effects below this inhibitory threshold in order to boost NKT activation.
- In paragraph “Brain Cancer”, page 3, line 130, the reference 44 in the group 44-46 should be deleted since in this sentence the authors referred to the decrease of NKT cells in advanced solid tumors;
Thank you for identifying our error. We have corrected our mistake by removing reference 44 in line 130, since it presents data on NKT cells in glioma patients, whereas 45-46 represent papers with contrasting data on NKT cell numbers in other advanced solid tumor types.
- In same paragraph, the authors incorrectly reported studies on Neuroblastoma which is not a brain cancer but is an extracranial tumor. So the paragraph tittle should be correct as Brain Cancers and Neuroblastoma;
Thank you, we recognize our error and have changed the section title to “Brain cancers and Neuroblastoma”.
- In the paragraph “Melanoma”, the first sentence in lines 150-154 is too long, unclear and needs a reference. Authors should rewrite these concepts. In line 158 “subset distribution” it is not clear which cell type this refers to;
Thank you for pointing out the lack of reference in line 147, we have added a reference (#52). Additionally, we agree with your suggestion that the sentence from lines 147-151 is unclear. We have corrected it to read, “Although a decrease in human NKT cells was observed in the peripheral blood of melanoma patients compared to healthy donors, patient NKT cells expanded similarly and mediated cytolytic activity against cell lines similar to healthy donors”.
We also thank you for your comment regarding line 158. That line refers an increase in CD8+ iNKT cell subset and simultaneous decrease in CD4+ iNKT cell subset in responders to treatment with MART-1-pulsed DCs. To provide additional detail as requested, we have added a sentence from lines 146-148 that reads, “Specifically, MART-1/DC responders experienced an increase in CD8+ iNKT cells and a concomitant decrease in CD4+ iNKT cells”.
- In paragraph “Colorectal cancer and Renal cell carcinoma” line 173 correct “significance difference” with” significant difference”; line 174 correct “compared against” with “ compare with”;
Thank you for pointing out these grammatical errors. We have made the appropriate changes to the sentence from lines 168-170.
- In the same paragraph, line 178, delete “in the patient population” as it is redundant.
Thank you for your suggestion, we have made the change accordingly in line 173.
- In line 179, authors refer to NK cells (?);
Thank you, we acknowledge our error and have changed “NK” to “NKT” cells.
- In line 182 “was attributed” referred to a-GalCer treatment is unclear and should be appropriately rewrite;
We apologize for the lack of clarity and have rewritten lines 169-173 to read, “ In murine models with colorectal metastases, treatment with α-GalCer significantly inhibited the progression of tumor growth and resulted in a high number of cured mice with acquired tumor-specific immunity”. We hope that the meaning of this sentence is no longer ambiguous.
- In lines 183-188 the sentences must be rewritten with care as they unclear and contain grammatical errors;
We have taken your comment into consideration and have changed the sentence from lines 173-174 to read, “Taken together with the studies focused on CD1d expression by tumors, this suggests a role for agents that induce CD1d-mediated antigen presentation as means of enhancing NKT cell mediated cancer immune surveillance in the tumor microenvironment”. We hope this now captures our intention to provide support for possible interventions focused on CD1d expression in tumors.
- In the paragraph “Other Cancer Types”, page 5, in line 211, add the explanation of the acronyms for MDSC by moving it from the line 222;
- In line 214 correct “can directly mediated” with “can directly mediate”;
Thank you for pointing out this grammatical error, we have made the suggested change.
- In line 215 correct “;” with “,”;
Thank you for the correction, we have replaced the semicolon with a comma.
- In lines 232-234 something seems to be missing after “found that CD1d+”…. carefully review the whole sentence;
- In line 273 review the sentence “this was provided further evidence”
Thank you for pointing out this grammatical error, we have made the appropriate change.
- On page 12, line 341, the authors should add a paragraphs of conclusions to summarize the meanings of the review, the description of therapeutic strategies referred to Figure 1 and add a Table with all current clinical trials on the use of NKT cells for solid tumors, by visiting clinicaltrial.gov website.
We think this is a wonderful suggestion. We have created Table 1 that contains relevant clinical trials involving NKT cells. We have added more details to the figure legend as well as to the text.